# Risk factors for multidrug-resistant and carbapenem-resistant *Klebsiella pneumoniae* bloodstream infections in Shanghai: A five-year retrospective cohort study

Hongwen Cao[1☯], Siqi Zhou[2☯], Xuefeng Wang[2], Shuzhen Xiao[2‡*], Shengyuan Zhao[3‡*]

**1** Department of Medical Laboratory, Xinyang Central Hospital, Xinyang, Henan, China, **2** Department of Laboratory Medicine, Ruijin Hospital, Shanghai Jiao Tong University School of Medicine, Shanghai, China, **3** Department of Clinical Laboratory, Xiangya Hospital, Central South University, Changsha, Hunan, China

☯ These authors contributed equally to this work.
‡ SX and SZ also contributed equally to this work.
* shengyuanzhao@csu.edu.cn (SYZ); zndxxsz@163.com (SZX)

## Abstract

### Background

Multidrug-resistant *Klebsiella pneumoniae* (MDRKP) and carbapenem-resistant *Klebsiella pneumoniae* (CRKP) bloodstream infections (BSIs) account for significant mortality and healthcare costs.

### Objectives

To investigate risk factors for MDRKP and CRKP BSIs

### Methods

A retrospective analysis of inpatients with *Klebsiella pneumoniae* bloodstream infections (KP BSIs) was conducted in a tertiary care hospital in Shanghai from 01/01/2018–31/12/2022. Temporal distribution of mortality and department distribution of KP BSIs were assessed. A generalized linear model was used to determine risk factors for MDRKP and CRKP BSIs.

### Results

A total of 379 inpatients with KP BSIs were included. The proportion of death for KP BSIs, MDRKP BSIs and CRKP BSIs gradually decreased since 2020. Majority of both MDRKP and CRKP BSIs patients were from Intensive Care Unit (ICU), burn unit, hematology and pancreatic surgery. Genitourinary disorders, invasive ventilator, history of antibiotic use, and carbapenem use were independently associated with MDRKP BSIs. Respiratory disease, gastric tubes, carbapenem use and its quantity were independently associated with CRKP BSIs.

**Data availability statement:** All relevant data are within the manuscript and its Supporting information files.

**Funding:** The author(s) received no specific funding for this work.

**Competing interests:** The authors have declared that no competing interests exist.

## Conclusions

ICU, burn unit, hematology and pancreatic surgery are common departments for MDRKP and CRKP BSIs. Genitourinary disorders, respiratory disorders, invasive ventilator, gastric tubes and antibiotic use (carbapenems in particular) within 90 days prior to onset of BSIs are independently associated with MDRKP and CRKP BSIs.

## Introduction

Bloodstream infections (BSIs) are strongly associated with increased morbidity and mortality, prolonged hospital stays, and high healthcare costs [1]. *Klebsiella pneumoniae* (KP) is the second most common cause of Gram-negative BSIs [2]. *Klebsiella pneumoniae* bloodstream infections (KP BSIs) is a clinical healthcare concern receiving significant attention worldwide. Incidence of KP BSIs has been gradually increasing in recent years [3]. Moreover, KP BSIs resulted in high mortality rates ranged between 20% and 40% [4]. In previous study, we found the 30-day crude mortality rate of KP BSIs in East China has reached 26.39% [5].

It has been reported in the literature that the drug-resistant phenotype of the pathogen is strongly associated with bad prognosis [6,7]. Our previous study also showed that mortality and medical costs of patients with MDRKP (multidrug-resistant *Klebsiella pneumoniae*) or CRKP (carbapenem-resistant *Klebsiella pneumoniae*) BSIs were significantly higher than those of patients with non-MDR/CR BSIs [5]. Moreover, the proportion of MDRKP BSIs and CRKP BSIs among KP BSIs in Shanghai was as high as 66.49% and 51.98%, respectively [5]. Therefore, investigating the risk factors for MDRKP and CRKP BSIs is important for the prevention and control of KP BSIs.

Previous studies have explored risk factors for CRKP BSIs while very few studies have reported risk factors for MDRKP BSIs. Therefore, it is necessary to update risk factors investigation for MDRKP and CRKP BSIs. For this reason, we retrospectively analyzed risk factors for patients with MDRKP and CRKP BSIs in a large tertiary hospital in East China with a high bacterial resistance rate.

## Materials and methods

### Study design and setting

We used medical record data between 01/01/2018 and 31/12/2022 abstracted from Hospital Information System of Ruijin Hospital which is developed by the Computer Center of Ruijin Hospital. These data were retrospectively assessed for research purpose from 30/09/2023–31/10/2023. Ruijin Hospital is a tertiary-level hospital with 3,697 beds in East China. Ruijin Hospital has an annual outpatient and emergency care volume of approximately 5.36 million, with a total of nearly 150,000 discharges per year. All consecutively hospitalized patients with KP BSIs during the study period were included. Patients with at least one result of KP in blood culture were included in our study. Only the first episode of KP BSIs per patient was included. Patients with the following conditions were excluded: (i) positive culture results considered as

contaminants; (ii) incomplete/inaccurate medical records; (iii) patients insisted on discharge from hospital against medical advice.

## Ethics approval and consent to participate

The study was approved by the ethics committee of Ruijin Hospital (No. KY2023–083) on 13/07/2023. This is an observational retrospective study and all the data were obtained from medical record systems and analyzed anonymously, so the committee waived informed consent. The study was conducted in accordance with the Declaration of Helsinki. We investigated incidence, antimicrobial resistance, crude 30-day mortality rate, and risk factors for crude 30-day mortality of KP BSIs using the same dataset of this manuscript (S1 File) and published the findings in 2024 [5]. In this study, we investigated temporal distribution of proportion of KP BSIs attributed deaths, department distribution of KP BSIs and risk factors for MDRKP BSIs and CRKP BSIs which are entirely different research questions. There is no overlap in research content of the two studies.

## Definitions

KP BSIs were defined as blood cultures positive for KP and accompanied by clinical signs and symptoms of infection [8]. The presence of KP BSIs was defined as the first positive blood culture of KP. Proportion of death for KP BSIs was defined as the number of KP BSIs attributed deaths per 100 patients with KP BSIs. Cases were defined as patients diagnosed with KP BSIs. MDR was defined as acquired insensitivity to at least one of three or more antimicrobial classes [9]. CR was defined as resistance to any carbapenems [10].

## Data collection

The following clinical information about the patient was extracted from the hospital's electronic medical record system: gender, age, department of admission, length of stay in ICU, length of hospitalization, underlying disease (malignancy, circulatory disease, endocrine disease, respiratory disease, genitourinary disease, gastrointestinal disease), history of invasive clinical procedures (surgery, paracentesis, arterial catheters, central venous catheters, invasive ventilators, urinary catheters, gastric catheters, drainage tubes), specific treatments (glucocorticosteroids, immunosuppressive agents, radiotherapy and chemotherapy), data on antibiotic therapy. Definitions of the variables collected are provided in S1 Table. All the datasets used and/or analyzed during the current study are listed in S1 File.

## Microbiology

The KP isolates were all characterized using MALDITOF MS (bioMérieux, Marcy l'Etoile, France), and drug sensitivity was tested using VITEK®2 (bioMérieux, Marcy l'Etoile, France). Interpretation of drug sensitivity tests was in accordance with the 2022 Clinical and Laboratory Standard Institute (CLSI) standard [11].

## Statistical analysis

Continuous variables that fit normal distribution were expressed as mean and standard deviation (SD), and continuous variables that did not fit normal distribution were expressed as median and interquartile range (IQR). Categorical variables were compared using the chi-square test and Fisher's exact test. A generalized linear model was used to determine risk factors for infection. In conducting the risk factor analysis, univariate analysis was performed first. Correlations and relevant interactions between variables with $P < 0.05$ in univariate analysis were examined. After excluding highly correlated variables (correlation coefficients ≥0.70), the remaining variables were considered for inclusion in the multiple logistic regression model and screened using the Lease Absolute Shrinkage and Selection Operator (LASSO) penalization method (used to select the variables with λ=lambda.1se, which is the cross-validation error plus a standard error minimized by λ) [12]. Selected variables

were included in the final multiple logistic regression model to determine their independent associations. The strength of these associations was determined by calculating odds ratio (OR) and 95% confidence interval (95% CI). All analyses were conducted using R version 4.2.1. To test the stability of the final multiple logistic regression model, variables were sequentially removed from the model and the significance of the remaining variables was checked [13]. Hosmer-Lemeshow test was performed to evaluate the goodness of fit of the regression models. $P<0.05$ was considered statistically significant.

## Results

### Trends in the proportion of death for MDRKP BSIs and CRKP BSIs

A total of 379 eligible subjects were included in the study. Among 379 patients with KP BSIs, there were 252 (66.5%) MDRKP cases and 197 (52.0%) CRKP cases. There was a total of 119 deaths during admission, with the proportion of death was 40.9% (103/252) in the MDR group, 12.6% (16/127) in the non-MDR group, 47.2% (93/197) in the CR group, and 14.3% (26/182) in the non-CR group, with a statistically significant difference between resistant group and non-resistant group (both $P<0.001$). The proportion of death for KP BSIs increased from 32.4% in 2018 to 36.0% in 2019, followed by a gradual decrease. The proportion of death for MDR group increased from 43.1% in 2018 to 52.2% in 2019, followed by a gradual decrease. The proportion of death for the non-MDR group gradually increased from 8.7% in 2018 to 16.2% in 2022. The proportion of death for the CR and non-CR groups increased from 48.7% and 14.3% in 2018 to 56.8% and 15.8% in 2019, respectively, and then gradually decreased. The details are shown in Fig 1.

### Distribution of departments

KP BSIs were predominantly distributed in ICU, pancreatic surgery, burn unit, and hematology (23.0%, 14.2%, 12.1%, and 10.8%, respectively, as shown in Fig 2A). MDRKP BSIs were predominantly distributed in the ICU (31.0%), followed by

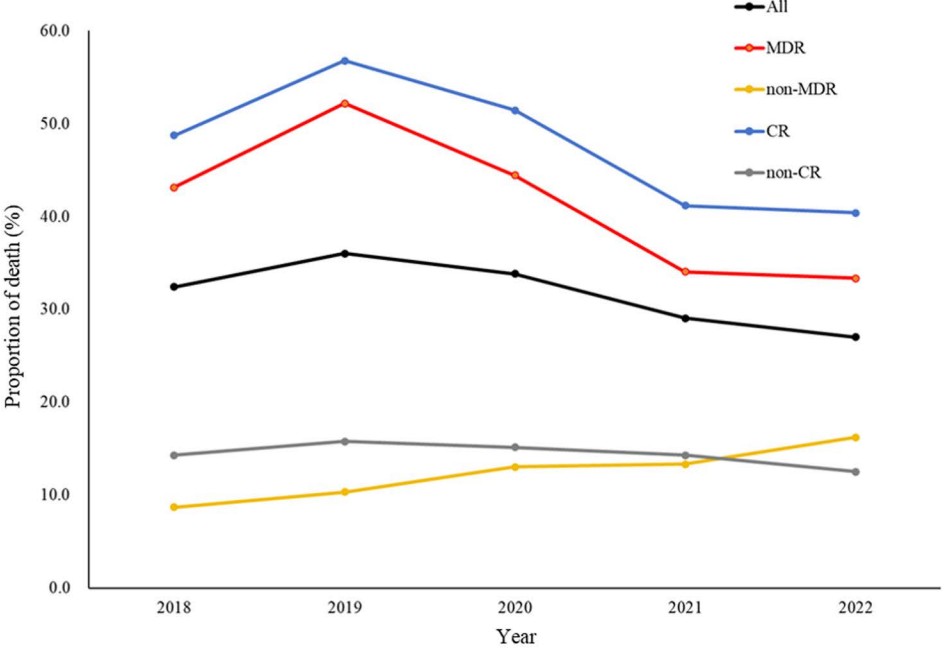

**Fig 1. Proportion of death among 379 inpatients with *Klebsiella pneumoniae* bloodstream infections (KP BSIs) from 2018 to 2022 by antimicrobial resistance phenotypes.** ALL, All KP BSIs; MDR, multidrug resistant; non-MDR, non-multidrug resistant; CR, carbapenem resistant; non-CR, non-carbapenem resistant.

burn unit (17.9%), hematology (11.5%), and pancreatic surgery (10.7%), details of which were shown in Fig 2B. Department distribution of CRKP BSIs was as well ICU (35.0%), followed by burn unit, hematology, and pancreatic surgery (22.3%, 11.2%, and 8.6%, respectively), as shown in detail in Fig 2C.

**Risk factors for MDRKP BSIs and CRKP BSIs**

Univariate analysis showed that age, male, some medical exposures within 90 days prior to the onset of BSIs, respiratory disease, genitourinary disease, gastrointestinal disease, some invasive procedures, use of some drugs within 90 days prior to diagnosis of BSIs were associated with MDRKP BSIs (Table 1). Multiple logistic regression analysis showed that genitourinary disease, invasive ventilator, history of antibiotic use in the 90 days prior to the diagnosis of BSIs, and carbapenem usage were independent risk factors for MDRKP BSIs (Table 1). The result of Hosmer-Lemeshow test ($\chi^2 = 8.78$, $P = 0.553$) was indicative of good fit. Univariate analyses showed that the factors associated with CRKP BSIs included age, male, some medical exposures within 90 days prior to the onset of BSIs, respiratory disease, gastrointestinal disease, some invasive procedures, use of some drugs within 90 days prior to diagnosis of BSIs (Table 2). Independent risk factors for CRKP BSIs were respiratory disease, indwelling gastric tube, and carbapenem use and quantity (Table 2). The result of Hosmer-Lemeshow test ($\chi^2 = 1.66$, $P = 0.948$) was indicative of good fit.

## Discussion

To the best of our knowledge, this is the first report of a risk factor analysis of both MDRKP and CRKP BSIs in eastern China, a region with a high prevalence of drug-resistant and virulent isolates [14]. The proportion of death for both MDRKP BSIs and CRKP BSIs rose to high point in 2019 and showed a gradual trend of decreasing afterwards, which we hypothesize is related to the fact that under China's coronavirus disease 2019 (COVID-19) policy, all patients who tested positive for nucleic acid were transferred to sentinel hospital in a closed loop. As China's COVID-19 control policy has been downgraded, more local studies should also be conducted in the future to analyze the impact of COVID-19 pandemics on mortality of BSIs.

Our study showed that the distribution of MDRKP and CRKP BSIs was highest in the ICU, which is consistent with other studies [15,16]. Interestingly, our risk factor analysis showed that ICU admission 90 days prior to diagnosis and the number of ICU days were risk factors for MDRKP and CRKP BSIs. The high prevalence of MDRKP and CRKP BSIs in the ICU maybe due to the presence of multiple infections in critically ill patients, a high number of invasive maneuvers, and frequent use of advanced antibiotic drugs. ICU has been described as factories that generate, spread and amplify antimicrobial resistance [1,17,18]. Selection of resistant strains and new mutations due to frequent and inappropriate application of antimicrobials influence emergence and rapid spread of multidrug-resistant pathogens in ICU [18]. Moreover, CRKP could be detected in various equipment of ICU including beds, tables, floors and ventilators, significantly higher than that of ordinary wards [1]. Many previous studies have shown that ICU admission was a risk factor for CRKP BSIs [15,19–22]. However, ICU admission 90 days before diagnosis was not an independent risk factor in the multiple logistic regression analysis, which may be influenced by other confounding factors. In addition, MDRKP and CRKP BSIs were higher in burn units and hematology. Burn patients are more likely to undergo invasive procedures and are more susceptible to infection, owing to burn wounds are favorable sites for bacterial colonization prior to healing [23]. Hematology inpatients with underlying hematologic malignancies, radiation and chemotherapy, neutropenia, gastrointestinal mucositis, and prolonged hospitalization are favorable conditions for the spread of MDRKP and CRKP BSIs [24].

As reported in other studies, age, male are risk factors for CRKP BSIs [1,16]. Our study found that age and male were risk factors for MDRKP BSIs. In addition, time at risk and length of hospitalization were risk factors for MDRKP and CRKP BSIs. The probability of bacterial infections developed from colonization rises as the length of hospitalization increases. Prolonged hospitalization has been reported to be a risk factor for colonization by antibiotic-resistant bacteria [25]. At the

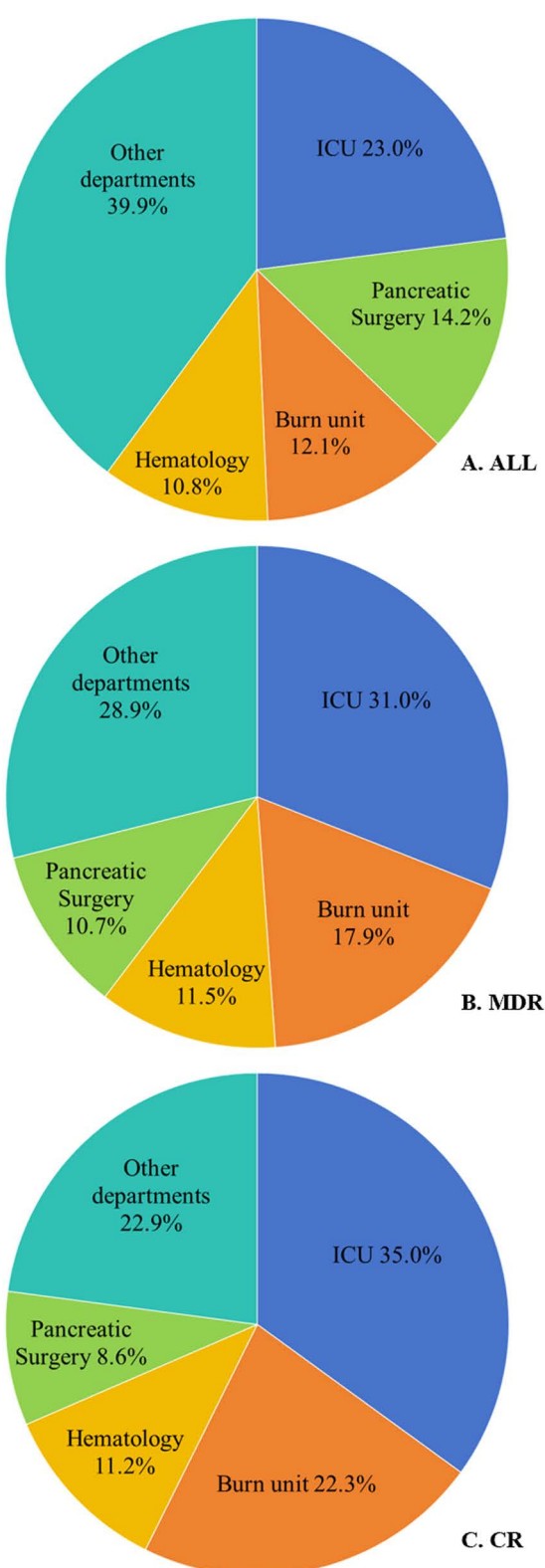

**Fig 2. Distribution of departments of 379 inpatients with *Klebsiella pneumoniae* bloodstream infections (KP BSIs) by antimicrobial resistance phenotypes.** ALL, all KP BSIs; MDR, multidrug resistant; CR, carbapenem resistant.

**Table 1. Risk factors for multidrug-resistant *Klebsiella pneumoniae* bloodstream infections (MDRKP BSIs).**

| Characteristics | MDR (%) (N = 252) | non-MDR (%) (N = 127) | Univariate | | Multiple | |
|---|---|---|---|---|---|---|
| | | | OR (95%CI) | P | aOR (95%CI) | P |
| Age (years), median (IQR) | 61.50 (49.00-70.00) | 64.00 (55.00-71.50) | 0.99 (0.97-1.00) | 0.042 | | |
| Gender, male | 189 (75.00) | 81 (63.78) | 1.70 (1.07-2.70) | 0.023 | | |
| Smoking | 45 (17.86) | 14 (11.02) | 1.75 (0.94-3.44) | 0.086 | | |
| Alcohol drinking | 37 (14.68) | 11 (8.66) | 1.81 (0.92-3.86) | 0.100 | | |
| Healthcare exposure | | | | | | |
| Time at risk (days), median (IQR) | 16.00 (7.00-31.00) | 6.00 (1.00-13.00) | 1.03 (1.02-1.05) | 0.000 | | |
| Length of hospital stay (days), median (IQR) | 19.00 (8.75-37.00) | 10.00 (3.00-21.50) | 1.02 (1.01-1.03) | 0.000 | | |
| ICU stay | 75 (29.76) | 17 (13.39) | 2.74 (1.57-5.02) | 0.000 | | |
| Length of ICU stay (days), median (IQR) | 0.00 (0.00-7.25) | 0.00 (0.00-0.00) | 1.02 (1.00-1.04) | 0.027 | | |
| Comorbidities | | | | | | |
| Chemotherapy or radiotherapy | 26 (10.32) | 17 (13.39) | 0.74 (0.39-1.45) | 0.375 | | |
| Malignancy | 49 (19.44) | 25 (19.69) | 0.98 (0.58-1.70) | 0.956 | | |
| Organic | 23 (9.13) | 12 (9.45) | 0.96 (0.47-2.06) | 0.919 | | |
| Hematology | 28 (11.11) | 13 (10.24) | 1.10 (0.56-2.26) | 0.796 | | |
| Disease of the circulatory system | 94 (37.30) | 44 (34.65) | 1.12 (0.72-1.76) | 0.612 | | |
| Hypertension | 77 (30.56) | 40 (31.50) | 0.96 (0.61-1.52) | 0.852 | | |
| Cerebrovascular disease | 12 (4.76) | 9 (7.09) | 0.66 (0.27-1.65) | 0.353 | | |
| IHD | 27 (10.71) | 10 (7.87) | 1.40 (0.68-3.14) | 0.381 | | |
| Endocrine, nutritional and metabolic diseases | 63 (25.00) | 35 (27.56) | 0.88 (0.54-1.43) | 0.591 | | |
| Diabetes mellitus | 47 (18.65) | 32 (25.20) | 0.68 (0.41-1.14) | 0.140 | | |
| Respiratory diseases | 35 (13.89) | 6 (4.72) | 3.25 (1.43-8.79) | 0.010 | | |
| Diseases of the genitourinary system | 36 (14.29) | 5 (3.94) | 4.07 (1.70-12.07) | 0.004 | 3.43 (1.27-11.06) | 0.023 |
| Diseases of the gastrointestinal system | 71 (28.17) | 53 (41.73) | 0.55 (0.35-0.86) | 0.008 | | |
| Invasive procedures | | | | | | |
| Surgery | 167 (66.27) | 70 (55.12) | 1.60 (1.03-2.48) | 0.035 | | |
| Paracentesis | 92 (36.51) | 20 (15.75) | 3.08 (1.82-5.41) | 0.000 | 1.89 (0.99-3.70) | 0.057 |
| AC | 52 (20.63) | 13 (10.24) | 2.28 (1.22-4.53) | 0.013 | | |
| Days of AC | 0.00 (0.00-0.00) | 0.00 (0.00-0.00) | 1.04 (1.00-1.09) | 0.090 | | |
| CVC | 181 (71.83) | 60 (47.24) | 2.85 (1.83-4.45) | 0.000 | | |
| Days of CVC | 7.00 (0.00-21.25) | 0.00 (0.00-5.50) | 1.03 (1.02-1.05) | 0.000 | | |
| Invasive ventilator | 110 (43.65) | 17 (13.39) | 5.01 (2.9-9.11) | 0.000 | 2.02 (1.02-4.12) | 0.047 |
| Days of indwelling invasive ventilator | 0.00 (0.00-7.00) | 0.00 (0.00-0.00) | 1.21 (1.11-1.34) | 0.000 | | |
| Urinary catheter | 158 (62.70) | 45 (35.43) | 3.06 (1.97-4.80) | 0.000 | | |
| Days of indwelling urinary catheter | 2.00 (0.00-15.00) | 0.00 (0.00-2.00) | 1.06 (1.03-1.09) | 0.000 | | |
| Gastric tube | 108 (42.86) | 19 (14.96) | 4.26 (2.51-7.55) | 0.000 | 1.38 (0.71-2.73) | 0.348 |
| Days of indwelling gastric tube | 0.00 (0.00-13.00) | 0.00 (0.00-0.00) | 1.05 (1.02-1.08) | 0.001 | | |
| Drainage tube | 130 (51.59) | 38 (29.92) | 2.50 (1.60-3.96) | 0.000 | | |
| Days of indwelling drainage tube | 1.00 (0.00-15.00) | 0.00 (0.00-1.50) | 1.03 (1.01-1.06) | 0.001 | | |
| Drug usage | | | | | | |
| Corticosteroids | 153 (60.71) | 35 (27.56) | 4.06 (2.57-6.52) | 0.000 | 1.41 (0.78-2.54) | 0.250 |
| Immunosuppressor | 24 (9.52) | 9 (7.09) | 1.38 (0.64-3.22) | 0.429 | | |
| Antibiotics | 223 (88.49) | 71 (55.91) | 6.07 (3.63-10.33) | 0.000 | 1.94 (1.00-3.80) | 0.049 |
| Combination therapy | 204 (80.95) | 44 (34.65) | 8.02 (4.99-13.1) | 0.000 | | |
| Glycopeptides | 117 (46.43) | 21 (16.54) | 4.37 (2.62-7.59) | 0.000 | 1.27 (0.62-2.60) | 0.514 |

*(Continued)*

**Table 1.** (Continued)

| Characteristics | MDR (%) (N = 252) | non-MDR (%) (N = 127) | Univariate | | Multiple | |
|---|---|---|---|---|---|---|
| | | | OR (95%CI) | P | aOR (95%CI) | P |
| Quantity (DDD), median (IQR) | 0.00 (0.00-5.00) | 0.00 (0.00-0.00) | 1.17 (1.09-1.28) | 0.000 | | |
| Vancomycin | 105 (41.67) | 19 (14.96) | 4.06 (2.39-7.20) | 0.000 | | |
| Quantity (DDD), median (IQR) | 0.00 (0.00-4.10) | 0.00 (0.00-0.00) | 1.17 (1.08-1.29) | 0.000 | | |
| Oxyazolidinones | | | | | | |
| Linezolid | 59 (23.41) | 5 (3.94) | 7.46 (3.2-21.81) | 0.000 | 2.61 (0.99-8.18) | 0.069 |
| Quantity (DDD), median (IQR) | 0.00 (0.00-0.00) | 0.00 (0.00-0.00) | 1.28 (1.14-1.53) | 0.001 | | |
| Carbapenems | 168 (66.67) | 32 (25.20) | 5.94 (3.71-9.69) | 0.000 | 1.23 (0.57-2.64) | 0.602 |
| Quantity (DDD), median (IQR) | 3.92 (0.00-13.00) | 0.00 (0.00-0.13) | 1.16 (1.10-1.23) | 0.000 | 1.05 (1.00-1.13) | 0.050 |
| Imipenem | 106 (42.06) | 27 (21.26) | 2.69 (1.66-4.47) | 0.000 | | |
| Quantity (DDD), median (IQR) | 0.00 (0.00-6.00) | 0.00 (0.00-0.00) | 1.10 (1.05-1.17) | 0.000 | | |
| Meropenem | 96 (38.10) | 8 (6.30) | 9.15 (4.54-21.13) | 0.000 | | |
| Quantity (DDD), median (IQR) | 0.00 (0.00-3.08) | 0.00 (0.00-0.00) | 1.30 (1.15-1.53) | 0.000 | | |
| Cephalosporins | 114 (45.24) | 36 (28.35) | 2.09 (1.33-3.33) | 0.002 | | |
| Quantity (DDD), median (IQR) | 0.00 (0.00-3.06) | 0.00 (0.00-1.00) | 1.05 (1.01-1.10) | 0.040 | | |
| β-lactam/β-lactamase inhibitor combinations | 32 (12.70) | 11 (8.66) | 1.53 (0.77-3.29) | 0.245 | | |
| Quantity (DDD), median (IQR) | 0.00 (0.00-0.00) | 0.00 (0.00-0.00) | 1.00 (0.97-1.04) | 0.921 | | |
| Piperacillin-tazobactam | 15 (5.95) | 6 (4.72) | 1.28 (0.50-3.65) | 0.623 | | |
| Quantity (DDD), median (IQR) | 0.00 (0.00-0.00) | 0.00 (0.00-0.00) | 0.99 (0.93-1.04) | 0.628 | | |
| Cefoperazone-sulbactam | 16 (6.35) | 5 (3.94) | 1.65 (0.63-5.15) | 0.337 | | |
| Quantity (DDD), median (IQR) | 0.00 (0.00-0.00) | 0.00 (0.00-0.00) | 1.02 (0.97-1.10) | 0.496 | | |
| Fluoroquinolones | 57 (22.62) | 10 (7.87) | 3.42 (1.75-7.35) | 0.007 | | |
| Quantity (DDD), median (IQR) | 0.00 (0.00-0.00) | 0.00 (0.00-0.00) | 1.09 (1.03-1.18) | 0.020 | | |

OR (95%CI), odds ratio (95% confidence interval); aOR (95%CI), adjusted odds ratio (95% confidence interval); IQR, interquartile range; BSIs, bloodstream infections; MDR, multidrug resistant; non-MDR, non-multidrug resistant; IHD, ischemic heart disease; ICU, intensive care unit; AC, arterial catheter; CVC, central venous catheter; DDD, defined daily dose.

same time, infections with drug-resistant bacteria make the patient's condition more complex and more difficult to treat, and ultimately lead to longer hospital stays.

Among the comorbidities, respiratory and genitourinary diseases were independent risk factors for CRKP BSIs and MDRKP BSIs, respectively. Most of the respiratory diseases are lung infections in this study (29/41, 70.7%), and previous studies have reported a relationship between lung infections and CRKP BSIs [16]. Common respiratory operations for patients with respiratory diseases, such as tracheal intubation, sputum suction, and tracheoscopy often lead to damage of airway mucosa, allowing the bacteria colonizing the respiratory tract or the equipment to enter the bloodstream, which can increase the probability of BSI [15,16]. Likewise, common invasive operations for patients with genitourinary diseases such as indwelling urinary catheter and urologic surgical procedures could damage the urinary tract and increase the risk of BSIs [15]. Additionally, our study found that surgical procedures and indwelling urinary catheter were strongly associated with MDRKP and CRKP BSIs (Tables 1 and 2). These operations tend to increase opportunities of infection by breaching the mucosal barrier, allowing bacteria to cross the barrier into the bloodstream [1,6,15].

Interestingly, both univariate and multiple logistic regression analyses indicated antibiotic exposure as a risk factor. Antibiotic exposure usually leads to the emergence of drug-resistant bacteria, and antibiotic use alters the microbiome, resulting in dominance of drug-resistant KP [24,26]. Carbapenems are usually considered as a last resort to combat infections caused by multidrug-resistant bacteria. Our results also confirmed that the use of carbapenems and its quantity

**Table 2. Risk factors for carbapenem-resistant *Klebsiella pneumoniae* bloodstream infections (CRKP BSIs).**

| Characteristics | CR (%) (N=197) | non-CR (%) (N=182) | Univariate OR (95%CI) | P | Multiple aOR (95%CI) | P |
|---|---|---|---|---|---|---|
| Age (years), median (IQR) | 60.00 (48.00-70.00) | 64.00 (55.00-72.00) | 0.99 (0.97-1.00) | 0.026 | | |
| Gender, male | 151 (76.65) | 119 (65.38) | 1.74 (1.11-2.73) | 0.016 | | |
| Smoking | 36 (18.27) | 23 (12.64) | 1.55 (0.88-2.76) | 0.132 | | |
| Alcohol drinking | 31 (15.74) | 17 (9.34) | 1.81 (0.98-3.47) | 0.064 | | |
| Healthcare exposure | | | | | | |
| Time at risk (days), median (IQR) | 17.00 (8.00-31.00) | 7.00 (2.00-18.00) | 1.01 (1.00-1.02) | 0.011 | | |
| Length of hospital stay (days), median (IQR) | 20.00 (9.00-37.00) | 12.00 (3.00-26.75) | 1.01 (1.00-1.02) | 0.008 | | |
| ICU stay | 69 (35.03) | 23 (12.64) | 3.73 (2.23-6.42) | 0.000 | 1.45 (0.73-2.89) | 0.288 |
| Length of ICU stay (days), median (IQR) | 0.00 (0.00-10.00) | 0.00 (0.00-0.00) | 1.02 (1.01-1.04) | 0.007 | | |
| Comorbidities | | | | | | |
| Chemotherapy or radiotherapy | 19 (9.64) | 24 (13.19) | 0.70 (0.37-1.33) | 0.279 | | |
| Malignancy | 31 (15.74) | 43 (23.63) | 0.60 (0.36-1.01) | 0.054 | | |
| Organic | 13 (6.60) | 22 (12.09) | 0.51 (0.24-1.04) | 0.069 | | |
| Hematology | 18 (9.14) | 23 (12.64) | 0.70 (0.36-1.33) | 0.275 | | |
| Disease of the circulatory system | 77 (39.09) | 61 (33.52) | 1.27 (0.84-1.94) | 0.261 | | |
| Hypertension | 65 (32.99) | 52 (28.57) | 1.23 (0.80-1.91) | 0.352 | | |
| Cerebrovascular disease | 9 (4.57) | 12 (6.59) | 0.68 (0.27-1.64) | 0.392 | | |
| Heart failure | 7 (3.55) | 7 (3.85) | 0.92 (0.31-2.74) | 0.880 | | |
| IHD | 18 (9.14) | 19 (10.44) | 0.86 (0.43-1.71) | 0.670 | | |
| Endocrine, nutritional and metabolic diseases | 50 (25.38) | 48 (26.37) | 0.95 (0.60-1.51) | 0.825 | | |
| Diabetes mellitus | 38 (19.29) | 41 (22.53) | 0.82 (0.50-1.35) | 0.438 | | |
| Respiratory diseases | 32 (16.24) | 9 (4.95) | 3.73 (1.80-8.52) | 0.001 | 3.39 (1.40-8.78) | 0.009 |
| Diseases of the genitourinary system | 24 (12.18) | 17 (9.34) | 1.35 (0.70-2.63) | 0.375 | | |
| Diseases of the gastrointestinal system | 52 (26.40) | 72 (39.56) | 0.55 (0.35-0.84) | 0.007 | | |
| Invasive procedures | | | | | | |
| Surgery | 132 (67.01) | 105 (57.69) | 1.49 (0.98-2.27) | 0.062 | | |
| Paracentesis | 71 (36.04) | 41 (22.53) | 1.94 (1.24-3.07) | 0.004 | | |
| AC | 46 (23.35) | 19 (10.44) | 2.61 (1.49-4.75) | 0.001 | | |
| Days of AC | 0.00 (0.00-0.00) | 0.00 (0.00-0.00) | 1.03 (1.00-1.07) | 0.085 | | |
| CVC | 146 (74.11) | 95 (52.20) | 2.62 (1.71-4.06) | 0.000 | | |
| Days of CVC | 8.00 (0.00-22.00) | 1.00 (0.00-10.00) | 1.01 (1.00-1.03) | 0.007 | | |
| Invasive ventilator | 100 (50.76) | 27 (14.84) | 5.92 (3.65-9.85) | 0.000 | 1.82 (0.95-3.47) | 0.069 |
| Days of indwelling invasive ventilator | 1.00 (0.00-9.00) | 0.00 (0.00-0.00) | 1.13 (1.08-1.19) | 0.000 | | |
| Urinary catheter | 135 (68.53) | 68 (37.36) | 3.65 (2.40-5.61) | 0.000 | | |
| Days of indwelling urinary catheter | 5.00 (0.00-17.00) | 0.00 (0.00-2.00) | 1.05 (1.03-1.08) | 0.000 | | |
| Gastric tube | 98 (49.75) | 29 (15.93) | 5.22 (3.25-8.60) | 0.000 | 2.12 (1.15-3.90) | 0.016 |
| Days of indwelling gastric tube | 0.00 (0.00-15.00) | 0.00 (0.00-0.00) | 1.06 (1.04-1.09) | 0.000 | | |
| Drainage tube | 108 (54.82) | 60 (32.97) | 2.47 (1.63-3.76) | 0.000 | | |
| Days of indwelling drainage tube | 1.00 (0.00-16.00) | 0.00 (0.00-2.00) | 1.03 (1.02-1.05) | 0.000 | | |
| Drug usage | | | | | | |
| Corticosteroids | 128 (64.97) | 60 (32.97) | 3.77 (2.48-5.80) | 0.000 | 1.36 (0.78-2.36) | 0.276 |
| Immunosuppressor | 15 (7.61) | 18 (9.89) | 0.75 (0.36-1.54) | 0.434 | | |
| Antibiotics | 184 (93.40) | 110 (60.44) | 9.26 (5.06-18.22) | 0.000 | | |
| Combination therapy | 173 (87.82) | 75 (41.21) | 10.28(6.21-17.60) | 0.000 | | |

*(Continued)*

**Table 2.** (Continued)

| Characteristics | CR (%) (N = 197) | non-CR (%) (N = 182) | Univariate | | Multiple | |
|---|---|---|---|---|---|---|
| | | | OR (95%CI) | *P* | aOR (95%CI) | *P* |
| Glycopeptides | 102 (51.78) | 36 (19.78) | 4.35 (2.77-6.96) | 0.000 | | |
| Quantity (DDD), median (IQR) | 0.25 (0.00-6.00) | 0.00 (0.00-0.00) | 1.09 (1.04-1.14) | 0.000 | | |
| Vancomycin | 91 (46.19) | 33 (18.13) | 3.88 (2.44-6.27) | 0.000 | | |
| Quantity (DDD), median (IQR) | 0.00 (0.00-5.00) | 0.00 (0.00-0.00) | 1.07 (1.03-1.13) | 0.001 | | |
| Oxyazolidinones | | | | | | |
| Linezolid | 54 (27.41) | 10 (5.49) | 6.50 (3.33-13.96) | 0.000 | 2.17 (0.98-5.08) | 0.063 |
| Quantity (DDD), median (IQR) | 0.00 (0.00-1.50) | 0.00 (0.00-0.00) | 1.25 (1.14-1.41) | 0.000 | | |
| Aminoglycosides | 26 (13.20) | 5 (2.75) | 5.38 (2.19-16.20) | 0.001 | | |
| Quantity (DDD), median (IQR) | 0.00 (0.00-0.00) | 0.00 (0.00-0.00) | 1.19 (1.06-1.46) | 0.033 | | |
| Carbapenems | 151 (76.65) | 49 (26.92) | 8.91 (5.64-14.32) | 0.000 | 2.74 (1.43-5.23) | 0.002 |
| Quantity (DDD), median (IQR) | 6.00 (0.25-15.00) | 0.00 (0.00-0.75) | 1.17 (1.13-1.23) | 0.000 | 1.06 (1.01-1.12) | 0.018 |
| Imipenem | 96 (48.73) | 37 (20.33) | 3.72 (2.38-5.93) | 0.000 | | |
| Quantity (DDD), median (IQR) | 0.00 (0.00-9.00) | 0.00 (0.00-0.00) | 1.15 (1.09-1.22) | 0.000 | | |
| Meropenem | 87 (44.16) | 17 (9.34) | 7.68 (4.43-14.01) | 0.000 | | |
| Quantity (DDD), median (IQR) | 0.00 (0.00-4.50) | 0.00 (0.00-0.00) | 1.17 (1.10-1.26) | 0.000 | | |
| Cephalosporins | 93 (47.21) | 57 (31.32) | 1.96 (1.29-3.00) | 0.002 | | |
| Quantity (DDD), median (IQR) | 0.00 (0.00-4.00) | 0.00 (0.00-1.00) | 1.06 (1.02-1.10) | 0.008 | | |
| β-lactam/β-lactamase inhibitor combinations | 25 (12.69) | 18 (9.89) | 1.32 (0.70-2.55) | 0.392 | | |
| Quantity (DDD), median (IQR) | 0.00 (0.00-0.00) | 0.00 (0.00-0.00) | 1.00 (0.97-1.03) | 0.930 | | |
| Piperacillin-tazobactam | 10 (5.08) | 11 (6.04) | 0.83 (0.34-2.02) | 0.681 | | |
| Quantity (DDD), median (IQR) | 0.00 (0.00-0.00) | 0.00 (0.00-0.00) | 0.98 (0.91-1.03) | 0.503 | | |
| Cefoperazone-sulbactam | 13 (6.60) | 8 (4.40) | 1.54 (0.63-3.97) | 0.352 | | |
| Quantity (DDD), median (IQR) | 0.00 (0.00-0.00) | 0.00 (0.00-0.00) | 1.01 (0.97-1.07) | 0.579 | | |
| Fluoroquinolones | 45 (22.84) | 22 (12.09) | 2.15 (1.25-3.81) | 0.007 | | |
| Quantity (DDD), median (IQR) | 0.00 (0.00-0.00) | 0.00 (0.00-0.00) | 1.01 (0.98-1.04) | 0.605 | | |

OR (95%CI), odds ratio (95% confidence interval); aOR (95%CI), adjusted odds ratio (95% confidence interval); IQR, interquartile range; BSIs, bloodstream infections; CR, carbapenem resistant; non-CR, non-carbapenem resistant; IHD, ischemic heart disease; ICU, intensive care unit; AC, arterial catheter; CVC, central venous catheter; DDD, defined daily dose.

prior to BSIs were independent risk factors for CRKP BSIs. Carbapenem use induces the production of acquired *Klebsiella pneumoniae* carbapenemase (KPC), which is one of the main mechanisms of CRKP resistance [27]. Moreover, CRKP can be transmitted by drug-resistant plasmids, which can cause more widespread CRKP infections [28]. Similar to some previous reports [29,30], we also found that prior exposure to quinolones was also an important risk factor. This may be due to the fact that the plasmid-encoded quinolone resistance determinant gene is located on the KP plasmid containing the KPC gene [31]. Additionally, quinolone use may have reduced Oprd pore protein expression and caused upregulation of the multidrug efflux pump MexEF-OprN, which is also a mechanism of carbapenem resistance [32]. We identified glucocorticoids as a risk factor for CRKP BSIs and MDRKP BSIs, however, multiple logistic regression analysis indicated that they were not an independent predictor. Glucocorticoids are recognized as a marker of an immunocompromised state and are more susceptible to serious infections [33]. Unlike other studies, we found that combination antimicrobial therapy may be a risk factor for MDRKP and CRKP BSIs, and we hypothesize that this is because combination therapy increases bacterial resistance to antibiotics [34]. In addition, we found that previous use of glycopeptides and cephalosporins were risk factors for MDRKP and CRKP BSIs, which is consistent with other studies [1,6]. We found that prior use of linezolid may increase the risk of MDRKP and CRKP BSIs, although multifactorial analyses showed that it was not an independent risk

factor. This may be due to the fact that most of the patients with prior linezolid use in our study were concurrently using carbapenems and thus would have been skewed. And whether prior use of linezolid directly leads to increased infections with drug-resistant bacteria needs to be further investigated in the future.

This study has several limitations. First, this is a single-center retrospective investigation with a relatively limited sample size which may not fully represent the characteristics of inpatients from other regions in China. In the future, a large-scale prospective multicenter study is needed to validate the results and accordingly to enhance the generalizability and reliability of the findings. Second, data on environmental exposure such as contamination of KP on the surfaces of objects like hospital bed rail and stethoscope, and inpatient density of wards were not available. These factors might have had an impact on risk factors for MDRKP and CRKP BSIs. Future research should address these points. Additionally, the mechanisms of CRKP and MDRKP are complex and diverse, and the risk factors may vary with different resistant mechanisms. This study did not detect resistant mechanisms, and molecular and bioinformatics studies should be conducted in future studies to better understand the effect of resistance mechanisms on infections.

Our study sheds light on which inpatients are at high risk of acquiring MDRKP and CRKP BSIs. This may have important implications for clinical interventions and public health policies. First, the findings would guide clinicians to identify inpatients at high-risk of MDRKP and CRKP BSIs at an early stage and initiate prompt targeted infection control measures. Second, infection prevention strategies such as enhanced hand hygiene, environmental disinfection, and isolation protocols should be prioritized in the common departments of MDRKP and CRKP BSIs. Finally, the development of a screening or surveillance system for MDRKP and CRKP BSIs based on the findings might help to prevent and control MDRKP and CRKP BSIs.

## Conclusions

Genitourinary disease, invasive ventilator, history of antibiotic use (carbapenems in particular) in the 90 days prior to the diagnosis of BSIs are independent risk factors for MDRKP BSIs, whereas independent risk factors for CRKP BSIs include respiratory disease, gastric tubes, and carbapenem use in the 90 days prior to the diagnosis of BSIs. ICU, burn unit, hematology and pancreatic surgery are common departments of MDRKP and CRKP BSIs. Pre-emptive identification and isolation measures among inpatients with these risk factors or from these departments would be a cost-effective way to identify, manage, and reduce the spread of MDRKP and CRKP BSIs.

## Supporting information

**S1 Table. Definitions of each variable included in risk factors analysis for multidrug-resistant *Klebsiella pneumoniae* and carbapenem-resistant *Klebsiella pneumoniae* bloodstream infections.**
(DOCX)

**S1 File. Dataset of clinical characteristics of 379 inpatients with *Klebsiella pneumoniae* bloodstream infections.**
(XLSX)

## Acknowledgments

We would like to thank Xuedong Wang for his assistance in the data organization process.

## Author contributions

**Conceptualization:** Shuzhen Xiao, Shengyuan Zhao.

**Data curation:** Hongwen Cao, Siqi Zhou.

**Formal analysis:** Hongwen Cao, Siqi Zhou.

**Investigation:** Hongwen Cao, Siqi Zhou.

**Methodology:** Hongwen Cao, Siqi Zhou.

**Project administration:** Shuzhen Xiao, Shengyuan Zhao.

**Resources:** Xuefeng Wang, Shuzhen Xiao, Shengyuan Zhao.

**Software:** Hongwen Cao, Siqi Zhou.

**Supervision:** Shuzhen Xiao, Shengyuan Zhao.

**Validation:** Xuefeng Wang.

**Visualization:** Hongwen Cao, Siqi Zhou.

**Writing – original draft:** Hongwen Cao, Siqi Zhou.

**Writing – review & editing:** Xuefeng Wang, Shuzhen Xiao, Shengyuan Zhao.

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
