## [Decision Letter · Decision Letter 0]

11 Feb 2025

PONE-D-24-58014Risk factors for multidrug-resistant and carbapenem-resistant Klebsiella pneumoniae bloodstream infections in Shanghai: a five-year retrospective cohort studyPLOS ONE

Dear Dr. Zhao,

Thank you for submitting your manuscript to PLOS ONE. After careful consideration, we feel that it has merit but does not fully meet PLOS ONE’s publication criteria as it currently stands. Therefore, we invite you to submit a revised version of the manuscript that addresses the points raised during the review process. Please submit your revised manuscript by Mar 28 2025 11:59PM. If you will need more time than this to complete your revisions, please reply to this message or contact the journal office at plosone@plos.org . Please include the following items when submitting your revised manuscript:

We look forward to receiving your revised manuscript.

Kind regards,

**Ali Amanati**

Academic Editor

PLOS ONE

2. In the online submission form, you indicated that [Some of relevant data are within the manuscript and its Supporting Information files. All the datasets used and/or analyzed during the current study are also available from the corresponding authors on reasonable request.].

Additional Editor Comments:

*
**Editor’s comments**
*

Your manuscript [PONE-D-24-58014] has passed the review stage and is ready for ‎revision. ‎To ensure the Editor and Reviewers can recommend that your revised manuscript be ‎accepted, ‎‎‎please pay careful attention to each comment posted underneath ‎this email. This way we ‎can ‎‎avoid future clarifications and revisions, moving swiftly to ‎a decision.‎

*
**Technical points:‎**
*

‎1. Please provide a point-by-point response to the Editor and reviewer's comments

‎2. Please highlight all the amends on your manuscript with a yellow color

‎3. Use line numbering and page number in the next submission‎

*
**The Editor’s main concern:‎**
*

Why wasn't there an investigation into the risk factors for mortality in Klebsiella pneumoniae bloodstream infections?

The study effectively explores CRKP and MDR KP. However, incorporating a discussion of risk scoring systems like the Increment CPE Score could strengthen the paper by providing a framework for predicting and understanding mortality risk in these infections. I recommend the authors consider adding this to the discussion.

Reviewers' comments:

Reviewer's Responses to Questions

**Comments to the Author**

1. Is the manuscript technically sound, and do the data support the conclusions?

Reviewer #1: Yes

Reviewer #2: Yes

Reviewer #3: Yes

Reviewer #4: Yes

2. Has the statistical analysis been performed appropriately and rigorously? 

Reviewer #1: Yes

Reviewer #2: Yes

Reviewer #3: Yes

Reviewer #4: Yes

3. Have the authors made all data underlying the findings in their manuscript fully available?

Reviewer #1: Yes

Reviewer #2: Yes

Reviewer #3: Yes

Reviewer #4: Yes

4. Is the manuscript presented in an intelligible fashion and written in standard English?

Reviewer #1: No

Reviewer #2: No

Reviewer #3: Yes

Reviewer #4: Yes

5. Review Comments to the Author

Reviewer #1: I appreciate the opportunity to evaluate this manuscript. It provides valuable insights into the risk factors for multidrug-resistant and carbapenem-resistant Klebsiella pneumoniae bloodstream infections, and I have a few comments and suggestions that may help enhance its clarity and impact.

The introduction contains several long sentences that could be broken down for better readability. For example, the sentence "Moreover, KP BSIs resulted in high mortality rates ranged between 20% and 40%, which has attracted attention" could be split into two sentences for clarity.

The terms "MDRKP" and "CRKP" are introduced without prior definitions in the background section.

The phrase "the resistance situation has become more serious in recent years" is somewhat redundant, as it implies a known trend without providing specific evidence.

The sentence "Only the first episode of each patient was included and only one episode per patient was included" is redundant. It can be simplify to: "Only the first episode of KP BSI per patient was included."

The phrasing "Definition of each variable corresponding to these data was listed in Appendix Table 1" is vague and should be made clearer. It can be stated as "Definitions of the variables collected are provided in Appendix Table 1."

The phrase "Because of the retrospective nature of the study, the committee waived informed consent" may lead to misinterpretation and could benefit from further clarification.

The discussion around ICU admissions is well-articulated, linking it to the increased prevalence of MDRKP and CRKP BSIs. However, it would strengthen the argument to include references or data that support the assertion that ICUs "generate, spread, and amplify antimicrobial resistance." Citing specific studies or statistics could bolster this claim.

The identification of respiratory and genitourinary diseases as independent risk factors is significant. Consider discussing the biological plausibility behind these associations in more detail, perhaps by referencing studies that have explored the mechanisms linking these conditions to increased susceptibility to infections.

To enhance the clarity and readability of your paper, I recommend avoiding nested parentheses (lines 96-97 and line 103).

In line 120, you mentioned the ratio of ratios (OR). It should be referred to as the Odds Ratio.

Have you used a generalized linear mixed model or a generalized linear model? (Line 110).

How did you assess the goodness of fit of the logistic regression model?

Multiple logistic regression is the more appropriate term than multivariate logistic regression.

Reviewer #2: Dear Authors

Thank for you the write-up. Kindly address the following concerns

General concerns

If patients with less than 48 hours of admission were excluded, then the title and focus of the study should reflect Hospital-acquired Klebsiella blood stream infections in order to properly align with the exclusion criteria otherwise, a clear and scientifically justifiable explanation should be given for the referenced exclusion criterion

The conclusion sounds too sentimental and would be better rephrased

specific concerns

Abstract should contain at least a statement of reflecting background before jumping to objectives

An abbreviation “CR” was introduced in the abstract which had not been previously explained

Line 51, readers will appreciate better if ‘good’ or ‘bad’ is used to properly qualify the ‘prognosis’ in the text.

Line 54% - please clarify is it proportion among all blood culture isolates or among Klebsiella isolates?

Line 57 – the import of the statement is not well articulated and difficult to understand

Line 120, kindly correct, ‘ratio of ratios’

Reviewer #3: Compared with some prior submissions, the writing in this article is exceptionally good. I read through it enough to get the sense that a fairly good writer composed this work, compared with the struggles one sometimes sees in articles that have to undergo repeated rewrites. This is a very useful topic in need of review across each health care system, and region with its own unique epidemiological, demographic and climatic-meteorologic-ecological settings. Long term studies focused on bacteriological phenotypic changes regarding MDRKP and CRKP treatment are most important, and very long term studies of changes in frequencies over time for a patter of ten to twenty years certain provides us with very important insights into the evolutionary process of pathogens and the development of drug resistance. This article is certain succinct enough for me with regard to how it reports the overall findings. Some individuals might be a little too detail oriented with figure 2, but that alone is not enough to really demand much of a change in the three (2 + 1) color patterns theoretically expressed due to the minor differences in figure A (ALL) versus the remaining two (a Cardiothoracic Surgery vs Pancreatic Surgery vs Burn Unit color mismatch.). This may very well be due to the printing of these figures and software issues, but I do not necessarily require that somehow this be fully and completely dealt with. In theory, such a correction or change should be quite easy to render in the submission of a final draft for publication. This change is requested, even suggested, but is not required.

Reviewer #4: Review Comments to the Author

The study is well-designed and addresses an important topic with significant clinical relevance, especially in regions with high levels of antimicrobial resistance. The statistical analysis is thorough, and the conclusions are well-supported by the data. The manuscript is written in clear English, and the figures and tables effectively illustrate the findings. However, the following recommendations are suggested to enhance the paper:

1. A more detailed discussion of the limitations is crucial for improving transparency and providing a clearer framework for future research.

2. Expanding on the implications for clinical interventions and public health policies would significantly increase the practical impact of the findings.

3. Additionally, the descriptive captions for figures and tables should be improved to ensure they are clear and easily understandable on their own.

4. The author should specify the name of the electronic medical record system used.

6. PLOS authors have the option to publish the peer review history of their article (what does this mean? ). If published, this will include your full peer review and any attached files.

**Do you want your identity to be public for this peer review?** For information about this choice, including consent withdrawal, please see our Privacy Policy .

Reviewer #1: **Yes: ** Farzane Ahmadi

Reviewer #2: No

Reviewer #3: **Yes: ** Brian L Altonen

Reviewer #4: No

---

## [Author Response · Author response to Decision Letter 1]

12 Mar 2025

Dear Editor,

We have submitted the manuscript entitled “Risk factors for multidrug-resistant and carbapenem-resistant Klebsiella pneumoniae bloodstream infections in Shanghai: a five-year retrospective cohort study” to PLOS ONE (manuscript ID: PONE-D-24-58014). Now we submitted a revised version of this manuscript, in which each point raised by editor and the reviewers has been addressed and all the changes have been highlighted in yellow throughout the manuscript. We deeply appreciate the valuable comments and suggestions.

Why wasn't there an investigation into the risk factors for mortality in Klebsiella pneumoniae bloodstream infections?

The study effectively explores CRKP and MDR KP. However, incorporating a discussion of risk scoring systems like the Increment CPE Score could strengthen the paper by providing a framework for predicting and understanding mortality risk in these infections. I recommend the authors consider adding this to the discussion.

Response: Thank you for your advice. Actually we have investigated risk factors for crude 30-day mortality of Klebsiella pneumoniae bloodstream infections (KP BSIs) using the same dataset of this manuscript and published the findings in 2024, please see reference 5 (https://doi.org/10.1080/23744235.2024.2374980). We found that MDR/CR phenotypes were associated with increased mortality of KP BSIs and healthcare costs (reference 5 and lines 50-55) and CR phenotype was one of independent risk factors for crude 30-day mortality of KP BSIs (reference 5). Therefore, we focused on exploring risk factors for MDRKP and CRKP BSIs in this manuscript. Although we reported in-hospital mortality rate in the current study, given that we have published the work on risk factors for crude 30-day mortality of KPBSIs using the same dataset, we did not look any further into risk factors for in-hospital mortality of KPBSIs in this study.

Thank you very much for your kind support and consideration of our manuscript.

Kind regards,

Shengyuan

Dear reviewers,

Thank you very much for your valuable comments and suggestions. We have responded to each of your comments and questions and revised the manuscript accordingly. We highlighted all the revisions throughout the manuscript in yellow. Please see our responses in the below.

Reviewer #1: I appreciate the opportunity to evaluate this manuscript. It provides valuable insights into the risk factors for multidrug-resistant and carbapenem-resistant Klebsiella pneumoniae bloodstream infections, and I have a few comments and suggestions that may help enhance its clarity and impact.

The introduction contains several long sentences that could be broken down for better readability. For example, the sentence "Moreover, KP BSIs resulted in high mortality rates ranged between 20% and 40%, which has attracted attention" could be split into two sentences for clarity.

Response: Thank you for your advice. We have double checked the background section and broken down several long sentences as suggested. Please see lines 41-42, 44-47, 50-55.

The terms "MDRKP" and "CRKP" are introduced without prior definitions in the background section.

Response: Apologies. We have added the definitions of “MDRKP” and “CRKP” as suggested. Please see lines 51-52.

The phrase "the resistance situation has become more serious in recent years" is somewhat redundant, as it implies a known trend without providing specific evidence.

Response: Apologies. You are right. We have deleted the phrase to make the sentence clear and meaningful. Please see lines 57-58.

The sentence "Only the first episode of each patient was included and only one episode per patient was included" is redundant. It can be simplify to: "Only the first episode of KP BSI per patient was included."

Response: Thank you for your advice. We have simplified the sentence as suggested to “Only the first episode of KP BSIs per patient was included.”, please see line 73.

The phrasing "Definition of each variable corresponding to these data was listed in Appendix Table 1" is vague and should be made clearer. It can be stated as "Definitions of the variables collected are provided in Appendix Table 1."

Response: Apologies. To make it clearer, we have rewritten the sentence as “Definitions of the variables collected are provided in S1 Table.” as suggested, please see lines 99-100.

The phrase "Because of the retrospective nature of the study, the committee waived informed consent" may lead to misinterpretation and could benefit from further clarification.

Response: Apologies. We have rewritten the sentence as “This is an observational retrospective study and all the data were obtained from medical record systems and analyzed anonymously, so the committee waived informed consent.” to make it clearer. Please see lines 79-81.

The discussion around ICU admissions is well-articulated, linking it to the increased prevalence of MDRKP and CRKP BSIs. However, it would strengthen the argument to include references or data that support the assertion that ICUs "generate, spread, and amplify antimicrobial resistance." Citing specific studies or statistics could bolster this claim.

Response: Thank you for your advice. We have added more supporting information and references as suggested, please see lines 202-206.

The identification of respiratory and genitourinary diseases as independent risk factors is significant. Consider discussing the biological plausibility behind these associations in more detail, perhaps by referencing studies that have explored the mechanisms linking these conditions to increased susceptibility to infections.

Response: Thank you for your advice. We have added more discussion and references as suggested, please see lines 227-236.

To enhance the clarity and readability of your paper, I recommend avoiding nested parentheses (lines 96-97 and line 103).

Response: Apologies. We have deleted unnecessary expressions to avoid nested parentheses as suggested, please see lines 96-98 and 104-105.

In line 120, you mentioned the ratio of ratios (OR). It should be referred to as the Odds Ratio.

Response: Apologies. We have corrected this mistake as suggested. Please see line 121.

Have you used a generalized linear mixed model or a generalized linear model? (Line 110).

Response: Apologies. We used a generalized linear model, and we have corrected this, please see line 111.

How did you assess the goodness of fit of the logistic regression model?

Response: Hosmer-Lemeshow test was performed to assess the goodness of fit of the logistic regression models. We have added this in the methods section (lines 125-126) and reported the results in results section (lines 170-171 and 176-177).

Multiple logistic regression is the more appropriate term than multivariate logistic regression.

Response: Thank you for your advice. We have changed to multiple logistic regression throughout the manuscript as suggested.

Reviewer #2: Dear Authors

Thank for you the write-up. Kindly address the following concerns

General concerns

If patients with less than 48 hours of admission were excluded, then the title and focus of the study should reflect Hospital-acquired Klebsiella blood stream infections in order to properly align with the exclusion criteria otherwise, a clear and scientifically justifiable explanation should be given for the referenced exclusion criterion

Response: In this study, the inclusion criteria for patients was primarily based on the results of blood cultures. Inpatients with at least one result of Klebsiella pneumoniae in blood culture were included in our study. According to the Centres for Disease Control and Prevention’s criteria, hospital-associated bloodstream infection (BSI) was defined as a BSI that occurred more than 48 hours after admission to hospital or within 48 hours of the last hospital discharge (doi:10.1016/j.ajic. 2008.03.002). For inpatients with more than 48 hours hospitalization but whose bloodstream infection occurred within 48 hours of hospital admission, we did not exclude them. In this study, there was one patient with schizophrenia who did not take the right amount of medicines on time as the physician suggested and was transferred to a psychiatric hospital for treatment 32 hours after hospital admission. This patient took the first positive blood culture of Klebsiella pneumoniae 20 hours after hospital admission. We could not know the exact amount of medicines the patient took by checking medical records. Therefore, we excluded this patient due to inaccurate medical records. There was one elder patient with advanced malignancy who was suggested to be admitted to ICU at admission given that the illness was very severe and the clinical prognosis would be quite poor without life support equipment. Both the patient and families refused to be transferred to ICU and insisted on discharge because they could not afford the healthcare costs for treatment. This patient took the first positive blood culture of Klebsiella pneumoniae at hospital admission. Therefore, we excluded this patient. The two patients mentioned above excluded from this study happened to be with less than 48 hours of hospitalization. As you suggested, the expression of “patients with less than 48 hours of hospitalization were excluded” is misleading, so we have rewritten the exclusion criteria to make it clearly justified as suggested, please see lines 75-76.

The conclusion sounds too sentimental and would be better rephrased

Response: Thank you for your advice. We have rephrased the conclusion section as suggested, please see lines 35-39 and 286-293.

specific concerns

Abstract should contain at least a statement of reflecting background before jumping to objectives

Response: Thank you for your advice. We have added the background statement as suggested, please see lines 19-21.

An abbreviation “CR” was introduced in the abstract which had not been previously explained

Response: Apologies. CR means CRKP BSIs, we have rewritten the sentence to make it clearer. Please see line 29.

Line 51, readers will appreciate better if ‘good’ or ‘bad’ is used to properly qualify the ‘prognosis’ in the text.

Response: Apologies. It should be bad prognosis. We have added the word to make it clear as suggested. Please see line 50.

Line 54 - please clarify is it proportion among all blood culture isolates or among Klebsiella isolates?

Response: Apologies. The proportion is among Klebsiella pneumoniae isolates. We have clarified this as suggested, please see line 54.

Line 57 – the import of the statement is not well articulated and difficult to understand

Response: Apologies. The phrase is somewhat redundant as it implies a known trend and not clear. We have rephrased the sentence to make it clear and meaningful. Please see lines 57-58.

Line 120, kindly correct, ‘ratio of ratios’

Response: Apologies. We have corrected this mistake as suggested. Please see line 121.

Reviewer #3: Compared with some prior submissions, the writing in this article is exceptionally good. I read through it enough to get the sense that a fairly good writer composed this work, compared with the struggles one sometimes sees in articles that have to undergo repeated rewrites. This is a very useful topic in need of review across each health care system, and region with its own unique epidemiological, demographic and climatic-meteorologic-ecological settings. Long term studies focused on bacteriological phenotypic changes regarding MDRKP and CRKP treatment are most important, and very long term studies of changes in frequencies over time for a patter of ten to twenty years certain provides us with very important insights into the evolutionary process of pathogens and the development of drug resistance. This article is certain succinct enough for me with regard to how it reports the overall findings. Some individuals might be a little too detail oriented with figure 2, but that alone is not enough to really demand much of a change in the three (2 + 1) color patterns theoretically expressed due to the minor differences in figure A (ALL) versus the remaining two (a Cardiothoracic Surgery vs Pancreatic Surgery vs Burn Unit color mismatch.). This may very well be due to the printing of these figures and software issues, but I do not necessarily require that somehow this be fully and completely dealt with. In theory, such a correction or change should be quite easy to render in the submission of a final draft for publication. This change is requested, even suggested, but is not required.

Response: Apologies. We have corrected the colors of Figure 2 to make sure that the colors match departments among the three groups as suggested.

Reviewer #4: Review Comments to the Author

The study is well-designed and addresses an important topic with significant clinical relevance, especially in regions with high levels of antimicrobial resistance. The statistical analysis is thorough, and the conclusions are well-supported by the data. The manuscript is written in clear English, and the figures and tables effectively illustrate the findings. However, the following recommendations are suggested to enhance the paper:

1. A more detailed discussion of the limitations is crucial for improving transparency and providing a clearer framework for future research.

Response: Thank you for your advice. We have added more detailed information about limitations in the discussion section as suggested, please see lines 264-275.

2. Expanding on the implications for clinical interventions and public health policies would significantly increase the practical impact of the findings.

Response: Thank you for your advice. We have added more discussion on the implications for clinical interventions and public health policies as suggested, please see lines 276-283.

3. Additionally, the descriptive captions for figures and tables should be improved to ensure they are clear and easily understandable on their own.

Response: Apologies. We have rephrased the descriptive captions for figures and tables to make them clear and easily understandable on their own as suggested.

4. The author should specify the name of the electronic medical record system used.

Response: Apologies. We have specified this as suggested, please see lines 65-67.

---

## [Decision Letter · Decision Letter 1]

24 Apr 2025

PONE-D-24-58014R1Risk factors for multidrug-resistant and carbapenem-resistant Klebsiella pneumoniae bloodstream infections in Shanghai: a five-year retrospective cohort studyPLOS ONE

Dear Dr. Zhao,

Thank you for submitting your manuscript to PLOS ONE. After careful consideration, we feel that it has merit but does not fully meet PLOS ONE’s publication criteria as it currently stands. Therefore, we invite you to submit a revised version of the manuscript that addresses the points raised during the review process. Please submit your revised manuscript by Jun 08 2025 11:59PM. If you will need more time than this to complete your revisions, please reply to this message or contact the journal office at plosone@plos.org . Please include the following items when submitting your revised manuscript:

We look forward to receiving your revised manuscript.

Kind regards,

**Ali Amanati**

Academic Editor

PLOS ONE

Journal Requirements:

Additional Editor Comments :

Dear authors,‎

We are pleased to inform you that your manuscript has successfully passed the ‎review stage and is ready for revision. Although the overall presentation of the ‎manuscript has improved following amendments, it still requires further ‎revision.‎

Yours sincerely,‎

Reviewers' comments:

Reviewer's Responses to Questions

**Comments to the Author**

1. If the authors have adequately addressed your comments raised in a previous round of review and you feel that this manuscript is now acceptable for publication, you may indicate that here to bypass the “Comments to the Author” section, enter your conflict of interest statement in the “Confidential to Editor” section, and submit your "Accept" recommendation.

Reviewer #1: All comments have been addressed

Reviewer #2: All comments have been addressed

2. Is the manuscript technically sound, and do the data support the conclusions?

Reviewer #1: Yes

Reviewer #2: Yes

3. Has the statistical analysis been performed appropriately and rigorously? 

Reviewer #1: Yes

Reviewer #2: Yes

4. Have the authors made all data underlying the findings in their manuscript fully available?

Reviewer #1: Yes

Reviewer #2: No

5. Is the manuscript presented in an intelligible fashion and written in standard English?

Reviewer #1: Yes

Reviewer #2: (No Response)

6. Review Comments to the Author

Reviewer #1: Dear authors,

Thank you for addressing all the comments in your manuscript. I have no further comments to add.

Reviewer #2: Dear authors

If the data set for this write-up has been used in previous publication (s), in order to conform to ethical standards, this must be clearly stated in the methodology section and the new or entirely different research question this write-up seeks to address unambiguously stated as well.

7. PLOS authors have the option to publish the peer review history of their article (what does this mean? ). If published, this will include your full peer review and any attached files.

**Do you want your identity to be public for this peer review?** For information about this choice, including consent withdrawal, please see our Privacy Policy .

Reviewer #1: **Yes: ** Farzane Ahmadi

Reviewer #2: No

---

## [Author Response · Author response to Decision Letter 2]

1 May 2025

Dear reviewer,

Thank you very much for your valuable comments and suggestions. We have responded to your comment and revised the manuscript accordingly. We highlighted all the revisions throughout the manuscript in yellow. Please see our responses in the below.

If the data set for this write-up has been used in previous publication (s), in order to conform to ethical standards, this must be clearly stated in the methodology section and the new or entirely different research question this write-up seeks to address unambiguously stated as well.

Response: Thank you. We have added the statement “We investigated incidence, antimicrobial resistance, crude 30-day mortality rate, and risk factors for crude 30-day mortality of KP BSIs using the same dataset of this manuscript (S1 File) and published the findings in 2024 [5]. In this study, we investigated temporal distribution of proportion of KP BSIs attributed deaths, department distribution of KP BSIs and risk factors for MDRKP BSIs and CRKP BSIs which are entirely different research questions. There is no overlap in research content of the two studies.” in the methodology section as suggested, please see lines 82-88.

---

## [Editor Report · Decision Letter 2]

4 May 2025

Risk factors for multidrug-resistant and carbapenem-resistant Klebsiella pneumoniae bloodstream infections in Shanghai: a five-year retrospective cohort study

PONE-D-24-58014R2

**
*Dear Dr. Shengyuan Zhao,*
**

We’re pleased to inform you that your manuscript has been judged scientifically suitable for publication and will be formally accepted for publication once it meets all outstanding technical requirements.

Kind regards,

Ali Amanati

Academic Editor

PLOS ONE

Additional Editor Comments (optional):

The authors have managed to use all the available resources and data to re-shape ‎the manuscript in a manner that is more scientifically sound than previously.‎

---

## [Editor Report · Acceptance letter]

PONE-D-24-58014R2

PLOS ONE

Dear Dr. Zhao,

I'm pleased to inform you that your manuscript has been deemed suitable for publication in PLOS ONE. Congratulations! Your manuscript is now being handed over to our production team.

Kind regards,

on behalf of

Professor Ali Amanati

Academic Editor

PLOS ONE